# Bizarre Morphology Obscures Real Affiliation: An Integrative Study of Enigmatic Cephalaspid *Philine denticulata* from Arctic Waters Reveals Its Unique Phylogenetic Position

Elena Chaban [1], Irina Ekimova [2], Pavel Lubin [3], Ekaterina Nikitenko [2] and Dimitry Schepetov [4,*]

1   Zoological Institute, Russian Academy of Sciences, Universitetskaya emb. 1, 199034 St. Petersburg, Russia
2   Invertebrate Zoology Department, Lomonosov Moscow State University, Leninskie gori 1-12, 199234 Moscow, Russia
3   State Budgetary Establishment Research Institute for Problems of Ecology and Mineral Wealth Use of Tatarstan Academy of Sciences, Daurskaya str. 28, 420087 Kazan, Russia
4   Biological Faculty, Shenzhen MSU-BIT University, Shenzhen 518172, China
*   Correspondence: d.m.schepetov@gmail.com

**Abstract:** The biodiversity of Cephalaspidea (Gastropoda: Heterobranchia) is poorly studied, and novel findings often lead to revisions at different taxonomic levels. The family Philinidae has a distinct set of defining characters in the shell, copulatory apparatus, and gizzard morphology, but several species, considered part of the family, deviate from it significantly. *Philine denticulata* (J. Adams, 1800) was considered to be a Philinidae despite the species morphology not fitting well with the family diagnosis. This species has an oval cylindrical external shell, jaws, and a gizzard lined with a chitinous layer with three thickened ridges. We studied *Philine denticulata* morphology on samples from the White and Barents Seas using the light and scanning electron microscopy methods in addition to computer microtomography. We also reconstructed its phylogenetic position using COI, 16S, 28S, and H3 genetic markers. Our integrative analysis revealed close relationships of this species to the family Aglajidae. Thus, we describe a new genus *Philinissima* gen. nov., including a single species *Philinissima denticulata* (J. Adams, 1800) comb. nov. which is the first aglajid registered from the Arctic waters. Our findings highlight possible problems in the Aglajidae taxonomical composition and an overall need for a dedicated integrative revision of the Cephalaspidea.

**Keywords:** Philinidae; Aglajidae; *Philinissima* gen. nov.; Gastropoda; Heterobranchia; morphology; taxonomy; ecology; phylogeny; Barents Sea; White Sea





## 1. Introduction

Marine biodiversity, especially in remote areas, receives only limited attention and is still poorly studied. Nevertheless, a considerable fraction of living organisms with taxonomic diversity can be exclusively found in seawater and brackish-water habitats. Better understanding of biodiversity is not only needed for setting proper conservation policies, but is also paramount for fundamental evolutionary and taxonomic research [1,2], and even partial data can significantly improve our understanding of high-level phylogeny [3]. A good representation of as many as possible lineages is essential for successful phylogenetic reconstruction [4–6], and the lack of sufficient data on some large, predominantly marine, groups severely hinder efforts to obtain a good resolution of the tree-of-life reconstruction at the levels of families and higher [7]. Even at a level as high as phyla new groups are still discovered. In the past half-century, three new phyla of animals were reported: Cycliophora were discovered less than 30 years ago [8], with Placozoa (see Grell [9]) and Loricifera (see Kristensen [10]) first found not long before that.

Among gastropod mollusks, the species-diverse heterobranch order Cephalaspidea remains one of the most poorly studied. During the past decade, the most attention in

studies on the cephalaspid taxonomy and phylogeny was given to the family Philinidae *s.l.* [11–19]. This was precipitated by their vast known morphological diversity and the advent of molecular approach to phylogenetics. As a result, the Philinidae *s.l.* was shown to be polyphyletic and was splitted into four family-level taxa: Philinidae *s.str.*; Laonidae Pruvot-Fol, 1954; Philinorbidae Oskars, Bouchet et Malaquias, 2015; and "Philinidae clade IV" (*sensu* Oskars et al. [14]) represented by a great number of undescribed species [14]. In addition, both morphological and molecular studies have shown Antarctic philinids to be significantly different from the rest of *Philine s.str.* For these taxa, two new genera [17,18] and a new family Antarctophilinidae Moles, Avila & Malaquias, 2019 were, therefore, introduced.

Philinidae *s.str.* and in particular the genus *Philine* are very diverse [12,16,18,19]; however, all of them are characterized by the presence of a muscular gizzard with calcified gizzard plates [14]. Amongst the philinid diversity, the external shell is only seen in three species: *Philine confusa* Ohnheiser, Malaquias, 2013; *Johania vestita* (Philippi, 1840) (Ohnheiser and Malaquias [12], as *J. retifera*), and *Philine denticulata* (J. Adams, 1800). At the same time, the lack of molecular data impedes the assessment of phylogenetic relationships of these species, leaving an open question whether the external shell is a synapomorphic or convergent trait. Moreover, *P. denticulata* morphology is inconsistent with contemporary *Philine s.str.* (and Philininae) diagnosis [14,18,19]: a muscular gizzard with calcified plates and a cephalic penial complex with a developed penial papilla, and separated incurrent and ejaculatory ducts. All stated above highlights the necessity of morphological redescription and molecular phylogenetic investigation of *P. denticulata*.

*Philine denticulata* is a widely distributed cephalaspid that was originally described from England, and its known areal encompasses the Northeast Atlantic from the Mediterranean Sea to Norway [12,20–25]. In addition, it was recently recorded as an abundant component of the Barents Sea fauna [26]. Since its initial description at the brink of the nineteenth century, *P. denticulata* was morphologically redescribed by Thompson [23] and later by Ohnheiser and Malaquias [12], while its reproduction, development, and life cycle were thoroughly studied by Horikoshi [24]. However, its phylogenetic position remains uncertain: no molecular data are available and no obvious morphological synapomorphies allow it to be grouped robustly and decisively with other taxa within Philinoidea.

The main goal of this study is an integrative taxonomic analysis of *P. denticulata* with a detailed morphological redescription using light microscopy and scanning electron microscopy (SEM) techniques, computer microtomography (mCT), and molecular phylogenetic analysis.

## 2. Material and Methods

### 2.1. Collection Data

Sample collection locations and details are listed in Table 1 and Figure 1A,B. A single specimen from the White Sea was collected in June 2015 from silted sand at Chupa Bay, cape Kartesh at upper sublittoral at depth of two meters. The sample of the muddy sediment was collected during scuba diving and then, immediately after collecting, sorted manually in the stationary laboratory using a stereomicroscope. The specimen was photographed in vivo and then was preserved in 96% EthOH and stored at −20 °C.

The newest samples from the White Sea were collected in the vicinity of the White Sea Biological Station of the Moscow State University in June 2022 using a Sigsbee trawl. The samples were washed in a gauze net and, immediately after collecting, sorted manually in the stationary laboratory using a stereomicroscope. Living specimens were kept at a temperature of +9 °C for study and then fixed in 96% EthOH. Two specimens were fixed in 2.5% glutaraldehyde in PBS for mCT analysis. The examined specimens have been deposited at the Zoological Institute, St. Petersburg, Russia (ZISP).

**Table 1.** Sites of sampling of the examined specimens of *Philinissima denticulata* comb. nov. from the White and Barents seas.

| Date | Expedition/Station/Collector | Coordinates | Depth (m) | Gear | Number of Specimens | Density (Specimens/m²) |
|---|---|---|---|---|---|---|
| 11 August 2003 | G/V Pomuald Muklevich, st. 18-3, P. Lubin | 69,5275 N 32,4511 E | 60 | Van Veen grab | 3 | 23 |
| 11 August 2003 | G/V Pomuald Muklevich, st. 18-4, P. Lubin | 69,5275 N 32,4511 E | 60 | Van Veen grab | 1 | |
| 23 August 2003 | G/V Pomuald Muklevich, st. 68-2, P. Lubin | 68,6804 N 41,6753 E | 70.6 | Van Veen grab | 1 | 7 |
| 23 August 2003 | G/V Pomuald Muklevich, st. 68-3, P. Lubin | 68,6804 N 41,6753 E | 70.6 | Van Veen grab | 1 | |
| 23 August 2003 | G/V Pomuald Muklevich, st. 69, P. Lubin | 69,0160 N 41,4670 E | 78.9 | Van Veen grab | 1 | 3 |
| 03 August 2008 | R/V Professor Boiko, st. TS1-1, P. Lubin | 69,1744 N 35,1509 E | 44.7 | Van Veen grab | 16 | 343 |
| 03 August 2008 | R/V Professor Boiko, st. TS1-2, P. Lubin | 69,1744 N 35,1509 E | 44.7 | Van Veen grab | 48 | |
| 03 August 2008 | R/V Professor Boiko, st. TS1-3, P. Lubin | 69,1744 N 35,1509 E | 44.7 | Van Veen grab | 39 | |
| 03 August 2008 | R/V Professor Boiko, st. TS2-1, P. Lubin | 69,1830 N 35,1437 E | 74 | Van Veen grab | 13 | 110 |
| 03 August 2008 | R/V Professor Boiko, st. TS2-2, P. Lubin | 69,1830 N 35,1437 E | 74 | Van Veen grab | 12 | |
| 03 August 2008 | R/V Professor Boiko, st. TS2-3, P. Lubin | 69,1830 N 35,1437 E | 74 | Van Veen grab | 8 | |
| 03 August 2008 | R/V Professor Boiko, st. TS4-1, P. Lubin | 69,2079 N 35,1652 E | 110 | Van Veen grab | 3 | 10 |
| 03 August 2008 | R/V Professor Boiko, st. TS5-1, P. Lubin | 69,2209 N 35,1606 E | 122.3 | Van Veen grab | 2 | 7 |
| 03 August 2008 | R/V Professor Boiko, st. TS7-1, P. Lubin | 69,2089 N 35,2224 E | 88.3 | Van Veen grab | 2 | 13 |
| 03 August 2008 | R/V Professor Boiko, st. TS7-3, P. Lubin | 69,2089 N 35,2224 E | 88.3 | Van Veen grab | 2 | |
| 03 August 2008 | R/V Professor Boiko, st. TS8-3, P. Lubin | 69,2076 N 35.2464 E | 63.3 | Van Veen grab | 5 | 17 |
| 03 August 2008 | R/V Professor Boiko, st. TS9-1, P. Lubin | 69,1996 N 35.2520 E | 58 | Van Veen grab | 2 | 13 |
| 03 August 2008 | R/V Professor Boiko, st. TS9-3, P. Lubin | 69,1996 N 35.2520 E | 58 | Van Veen grab | 2 | |
| 04 August 2008 | R/V Professor Boiko, st. TS11-2, P. Lubin | 69,2132 N 35,2678 E | −2 | Van Veen grab | 1 | 3 |
| 04 August 2008 | R/V Professor Boiko, st. TS12-2, P. Lubin | 69,2215 N 35,2282 E | −2 | Van Veen grab | 8 | 43 |
| 04 August 2008 | R/V Professor Boiko, st. TS12-3, P. Lubin | 69,2215 N 35,2282 E | −2 | Van Veen grab | 5 | |

**Table 1.** *Cont.*

| Date | Expedition/Station/Collector | Coordinates | Depth (m) | Gear | Number of Specimens | Density (Specimens/m$^2$) |
|---|---|---|---|---|---|---|
| 04 August 2008 | R/V Professor Boiko, st. TS13-1, P. Lubin | 69,2031 N 35,1952 E | 100.7 | Van Veen grab | 2 | |
| 04 August 2008 | R/V Professor Boiko, st. TS13-2, P. Lubin | 69,2031 N 35,1952 E | 100.7 | Van Veen grab | 2 | 20 |
| 04 August 2008 | R/V Professor Boiko, st. TS13-3, P. Lubin | 69,2031 N 35,1952 E | 100.7 | Van Veen grab | 2 | |
| June 2015 | A. Chaban, E. Chaban | White Sea, Kandalaksha Bay, Chupa Bay | 2 | scuba diving | 1 | |
| 20 June 2022 | A. Zhadan, E. Chaban | White Sea, Kandalaksha Bay, Ermolinskaya Bay | 8–9 | igsbee trawl | 18 | |

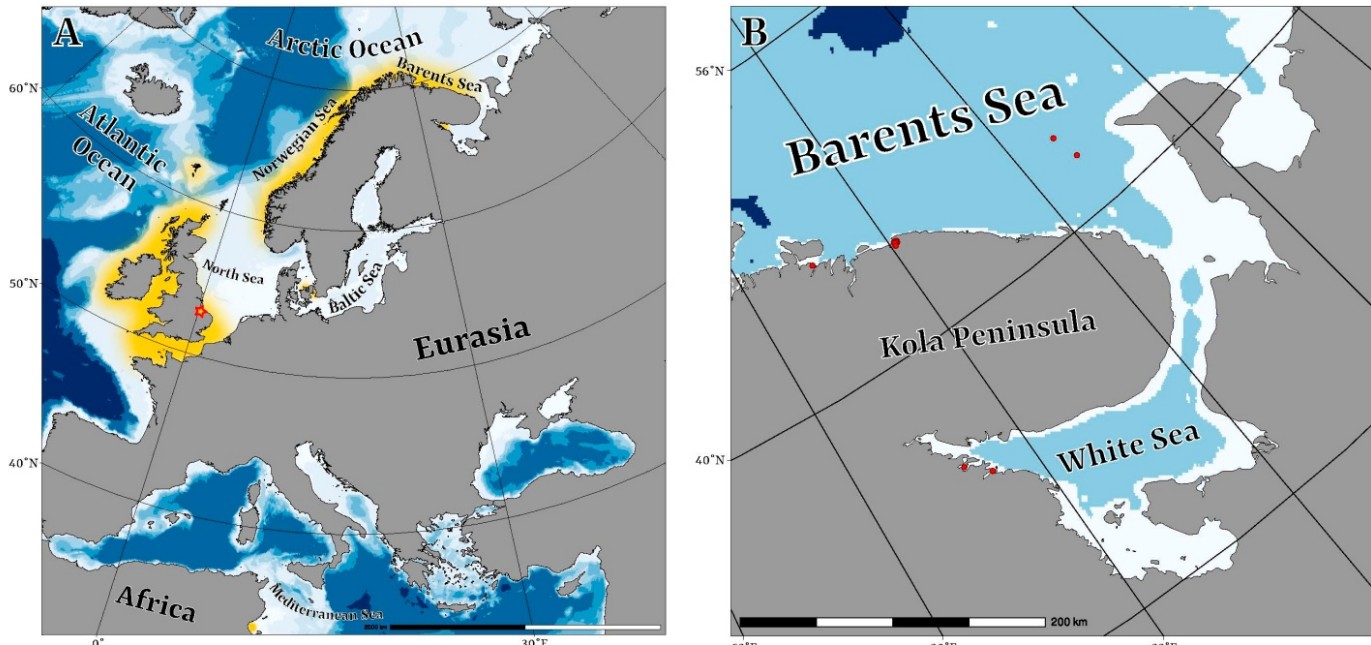

**Figure 1.** Maps showing: (**A**)—distribution of *Philinossima denticulata* in the world, (**B**)—collection sites of *P. denticulata* in the White and Barents Seas.

Specimens from the Barents Sea were collected during the expeditions of Murmansk Marine Biological Institute aboard the HSV Romuald Muklevitch in 2003 and RV Professor Boiko in 2008. The samples were collected with a 0.1 m$^2$ Van Veen grab sampler, fixated in 4% formaldehyde, and then sorted in the stationary laboratory and preserved in 70% EthOH.

### 2.2. DNA Extraction, PCR Amplification, and Sequencing

Genomic DNA was extracted from tissue samples stored in 96% EtOH (Table 2) following the CCDB invertebrate protocol [27]. Isolated DNA was used as a template for the amplification of the partial mitochondrial *cytochrome c oxidase* subunit I (COI), partial nuclear large ribosomal subunit RNA (28S), and the third histone subunit (H3) genetic markers. Although we failed to amplify the 16S rRNA marker for *P. denticulata* likely due to the variability of the binding site for standard 16S primers in this species, this locus was incorporated into the analysis to improve the resolution at the genus- and family-levels. The corresponding sites for *P. denticulata* were marked as missing data. The PCR and sequencing conditions, and the used primers followed previously published protocols [28]. Sequencing reactions were analyzed using an ABI 3500 Genetic Analyzer (Applied Biosystems). All novel sequences were submitted to the NCBI GenBank (Table 2).

**Table 2.** Specimens of *Philinissima denticulata* used for phylogenetic analyses.

| Species | Locality | DNA Extract No. | COI | 28S | H3 |
|---|---|---|---|---|---|
| *P. denticulata* | White Sea | EC333 | OQ445487 | OQ448320 | OQ453177 |
| *P. denticulata* | White Sea | EC334 | OQ445490 | OQ448321 | - |
| *P. denticulata* | White Sea | EC335 | - | OQ448322 | OQ453178 |
| *P. denticulata* | White Sea | EC336 | OQ445491 | OQ448323 | OQ453176 |
| *P. denticulata* | White Sea | EC337 | OQ445489 | OQ448324 | - |
| *P. denticulata* | White Sea | EC338 | - | - | OQ453175 |
| *P. denticulata* | White Sea | EC339 | OQ445488 | - | OQ453174 |
| *P. denticulata* | White Sea | EC340 | - | - | OQ453179 |

All raw reads for each marker were assembled and checked for ambiguities and low-quality data in Geneious R10 (Biomatters, Auckland, New Zealand). Edited sequences were verified for contamination using the BLAST-n algorithm run over the GenBank nr/nt database [29]. Original data and publicly available sequences were aligned with MUSCLE [30] algorithm in MEGA 7 [31]. Additionally, all protein-coding sequences were translated into amino acids to verify reading frames and check for stop codons.

### 2.3. Data Analysis

For the phylogenetic reconstruction, we used a molecular dataset obtained in previous study, which encompasses the current Philinoidea diversity [28] (Table S1). In order to test the phylogenetic position of *P. denticulata,* all currently accepted families of the superfamily Philinoidea (Aglajidae, Alacuppidae, Antarctophilinidae, Colpodaspididae, Gastropteridae, Laonidae, Philinidae, Philinoglossidae, Philinorbidae, and Scaphandridae) were added to the analysis. Members of families Retusidae and Cylichnidae were used as outgroups, *Diaphana globosa* was chosen as a distant outgroup following Oskars et al. [14]. Alignments for each marker comprised 658 bp for COI, 323 bp for 16S, 327 bp for H3, and 365 bp for 28S. Additionally, we checked protein coding genes for putative saturation by plotting uncorrected P-distances vs. TN93 model distances and checked how well the data fit the linear regression model (see Supplementary Figures S1 and S2). In case of COI alignment, where a moderate slope is observed, to mitigate possible problems caused by signal saturation [32], we ran an additional set of phylogeny inference with both methods, with the COI alignment separated into partitions by codon position.

Sequences were concatenated by a simple Biopython script published in Chaban et al. [19]. Phylogenetic reconstructions were performed for each marker and also for the concatenated multi-gene partitioned dataset. The resulting sequence alignment of concatenated COI, 16S, H3, and 28S markers contained 1673 positions. The best-fit nucleotide evolution models were tested in Modeltest-NG [33] under Bayesian Information Criterion (BIC) for each partition. Evolutionary models were applied separately to partitions representing single markers. The Bayesian phylogenetic analyses and estimation of posterior probabilities were performed in MrBayes 3.2 [34]. Markov chains were sampled at intervals of 500 generations. The analysis was initiated with a random starting tree and ran for $10^7$ generations. Run convergence was analyzed using Tracer 1.7.1 software [35]. Maximum-likelihood phylogeny inference was performed in the HPC-PTHREADS-AVX option of RaxML HPC-PTHREADS 8.2.12 [36] with the number of pseudoreplicates determined by the autoMRE algorithm under the GTRCAT model of nucleotide evolution. Final phylogenetic tree images were rendered in FigTree 1.4.0 and further modified in Adobe illustrator CS 2015.

### 2.4. Morphological Analysis

Living specimens were photographed in a Petri dish filled with unfiltered sea water using an Olympus PenPl1 and Leica DFC420 mounted on a Leica M 165 C stereomicroscope. The penial apparatus of the collected specimens was mounted in glycerol and examined under a light microscope Leica DME. Buccal masses and shells were cleaned in hypochlorite solution for 20 min. Then they were rinsed in distilled water, air-dried, mounted on an aluminum stub, and sputter-coated with gold and examined using the scanning electron microscopy (SEM; Quanta-250 and JCM 7000).

Two specimens were qualitatively studied by means of X-ray microtomography using a microtomograph SkyScan 1272 (Bruker MicroCT, Kontich, Belgium). The SkyScan 1272 settings were as follows: Al filter radiation, acceleration voltage—60 kV, current—166 µA, resolution—7.4 µm, crystal rotation angle—0.150 grad., 2 scans in one position, exposure—2823 msec. For the reconstruction of the shadow image array, a NRecon software package (BrukerMicroCT) was used, since it can neutralize the artefacts caused by the instrument and provide a grayscale range corresponding to X-ray absorption, and, consequently, to the chemical composition of the sample. Microtomographic sections obtained were analyzed using CTVox software packages.

*2.5. Nomenclatural Acts*

This published work and the nomenclatural acts it contains have been registered in the ZooBank, an online registration system for the ICZN. The ZooBank LSID (Life Science Identifier) for this publication is: urn:lsid:zoobank.org:pub:27378D89-3E36-4670-8FFE-6CE75E4D3BC8.

## 3. Results

*3.1. Systematics*

Class Gastropoda Cuvier, 1795
Subclass Heterobranchia Gray, 1840
Order Cephalaspidea Fischer, 1883
Superfamily Philinoidea Gray, 1850 (1815)
Family Aglajidae Pilsbry, 1895 (1847)
Genus *Philinissima* Chaban, Ekimova & Schepetov gen. nov.
ZooBank LSID: urn:lsid:zoobank.org:act:DCB93B8C-1B3B-412B-A7CB-BE5D16CBD383

*Diagnosis*: Shell external, white, cylindrical, partly covered by mantle, with fine net-like sculpture. Apex flat, consists of protoconch and 3/4 teleoconch whorl. Zone between protoconch and apical keel soft and with pores. Cephalic shield with oval or tapering posterior edge. Radula formula 1:1:1:1:1, jaws consist of rod-shaped elements. Chitinous lining of gizzard forming three zones with longitudinal ridges. Penis with shot prostate and conical unarmed penial papilla.

*Type species: Bulla denticulata* J. Adams, 1800.

*Comparison.* The new genus differs from all described genera of the family Aglajidae by the presence of an external ovate-cylindrical shell.

*Etymology.* The name *Philinissima* refers to *Philine* which is the base for all Philinacea.
*Philinissima denticulata* (J. Adams, 1800) comb. nov.
Figures 2–4.
*Bulla denticulata* J. Adams, 1800: 1, pl. 1, Figures 3–5.
*Type locality*: The Wash, England, UK.

*Material studied*: Barents Sea—24 lots (181 specimens); White Sea, Kandalaksha Bay—2 lots (19 specimens) (for collection data see Table 1).

*Diagnosis.* Corresponds to the genus diagnosis.
*Description. External morphology* (Figure 2): Body semi-transparent white or yellowish with small white pigment spots; length up to 3.8 mm. Cephalic shield of living specimen elongated with oval or tapering posterior edge. Small black eyes clearly visible. Hancock organs light brown bands along lateral edges of cephalic shield. Parapodia small; foot long, its posterior edge reaching shell apex. Shell mostly exposed; its anterior edge covered with mantle fold. Posteriorly mantle forming wide frill protecting apical part of shell. Larval kidney visible as small black spot though semi-transparent body and shell. Small gill also visible (Figure 2A) as yellow triangle on the right side of body at the middle part of shell. Gill apex pointing to the right.
*Shell* (Figures 2G,H and 3): external, white, cylindrical, up to 2.0 mm in length, consists of protoconch and 3/4 teleoconch whorl. Anterior edge of shell partly covered by mantle. Posteriorly mantle forming skirt on shell apex. Protoconch with 2 whorls (Figure 3E,H,I) but only 3/4 whorl visible in adult shell (Figure 3B). Protoconch clearly separated by bolster from teleoconch whorl. Internal zone of teleoconch whorl uncalcified, soft, with clearly visible rare pores (Figure 3D). Exposed protoconch in adult specimens with fine and rare axial folds. Protoconch of newly hatched larva consisting of 3/4 whorl with pit surface of its initial part (Figure 3G). It colorless, roundish-ovoid, with a wide aperture, 80 × 125 μm in its ventral aspect and 91 × 121 μm in lateral view (Figure 3H,I). Definitive whorl with net-like sculpture visible only in high magnification (Figure 3F). Callus wide on protoconch

and like narrow band on internal lip of aperture, forming so-called "wing-like" shape (Figure 3A).

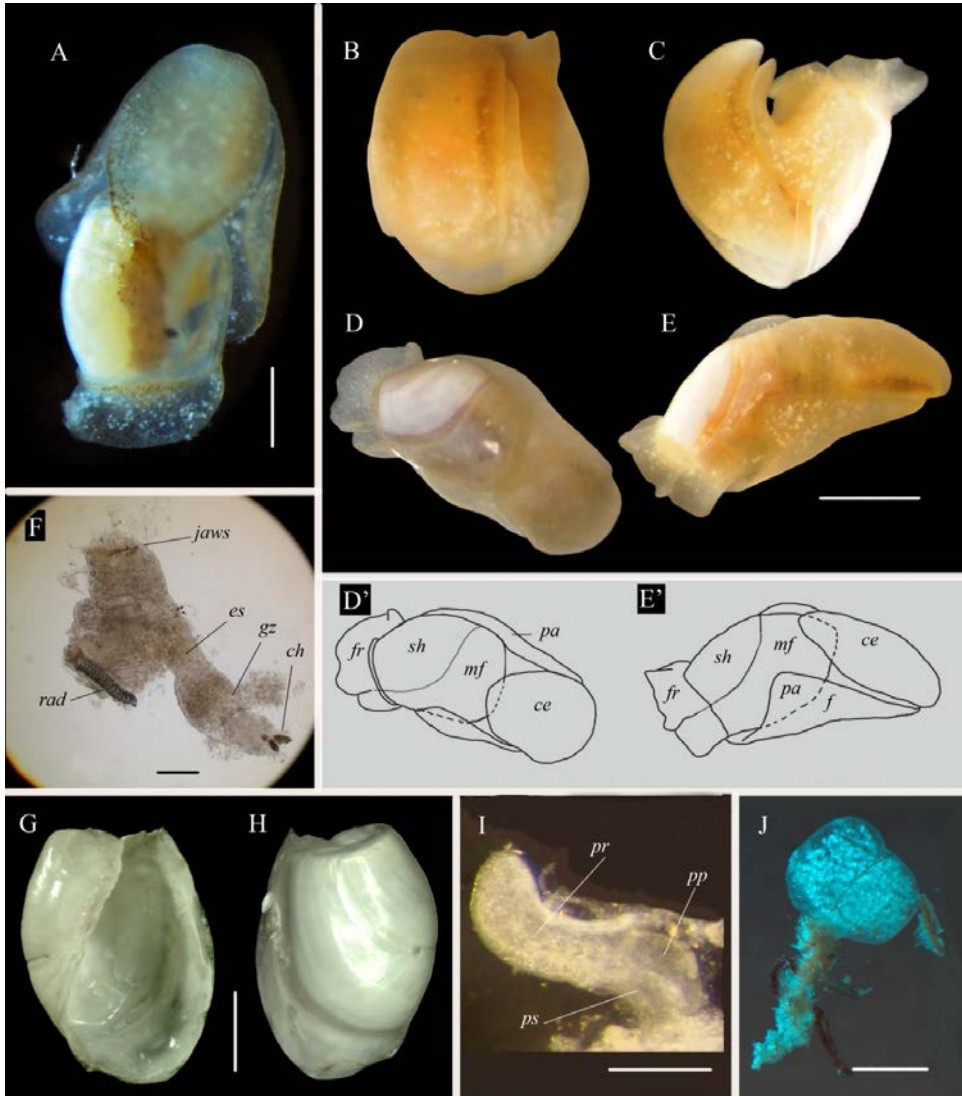

**Figure 2.** *Philinissima denticulata* specimens from the White Sea: (**A**,**D**)—dorsal view; (**B**–**E**)—moving specimen; (**E**)—lateral view; (**D′**,**E′**)—diagrams of figures (**D**) and (**E**) accordingly; (**B**,**C**), and (**D**) are the same specimen as (**E**); (**F**)—anterior part of digestive system; (**G**)—shell, ventral view; (**H**)—shell, dorsal view; (**I**)—copulatory system; (**J**)—egg mass of *P. denticulata* spawned in the aquarium. Abbreviations: ce—cephalic shield; ch—chitinous zones of gizzard, es—esophagus; f—foot; fr—mantle frill; gz—gizzard, mf—mantle fold; pa—parapodia, pp—penial papilla; pr—prostate; ps—penial sheath; rad—radula; sh—shell. Scale bars: 100 μm (**F**,**I**); 0.5 mm (**A**,**G**,**H**,**J**); 1 mm (**E**).

*Internal morphology* (Figures 2F,I and 4): Muscular pharynx bearing jaws anteriorly and radula posteriorly (Figure 2F). Jaws consisting of numerous rod-shaped elements (Figure 4G). Radula formula 15 × 1:1:1:1:1. Internal lateral tooth bearing small denticles on its internal edge (Figure 4F,I). Rachidian teeth consisting of two small wide wing-like plates with almost straight cutting edge and longitudinal crest (Figure 4H). Chitinous lining of gizzard forming three zone with longitudinal ridges (Figure 4A,D). Gonad continuing in visceral sac along median line of body. Testes brownish in color at the right side and ovaries yellow colored at the left side (Figure 2A). Hermaphrodite genital opening lying laterally at the right side presumably at the middle of the body (Figure 4B). From this point, external seminal groove visible between foot and cephalic shield up to cephalic copulatory system

opening (Figure 4B). Penis (Figure 2I) with short prostate (Figures 2I and 4E) and conical unarmed penial papilla.

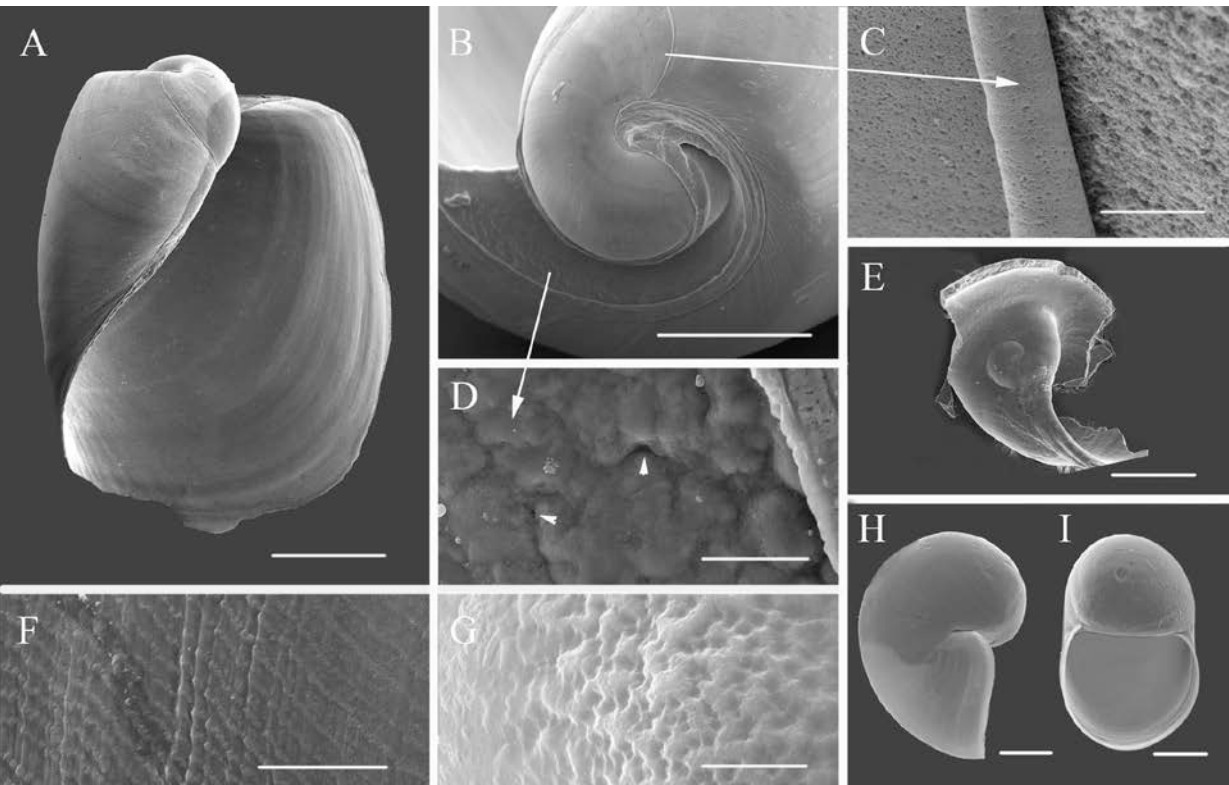

**Figure 3.** *Philinissima denticulata* collected in the Barents Sea, shell (SEM): (**A**)—ventral view; (**B**)—apical view; (**C**)—protoconch border; (**D**)—intercalated part of shell between protoconch and body whorl keel, pores marked by the short arrows; (**E**)—protoconch inside of broken definitive whorl; (**F**)—shell sculpture. Shell of newly hatched veliger: (**H**)—lateral view, (**I**)—ventral view, (**G**)—sculpture of veliger. Scale bars: 5 µm (**G**), 10 µm (**C,D**), 30 µm (**F,H,I**), 200 µm (**B,E**), and 500 µm (**A**).

*Distribution*. *Philinissima denticulata* has been recorded from many areas in the British Isles, Norway, and at one site of the Mediterranean Sea (see Horikoshi [24] for references and Figure 1A). Some authors considered the species as *Philine sinuata* Stimpson, 1850 or *Philinorbis sinuata* described from the West Atlantic (Boston Harbor) [21,37,38]. However, Franz and Clark [39] redescribed *Philine sinuata* and showed that the shell of this species is internal, and the formula of the radula is 2:1:0:1:2. Therefore, the areal of *P. denticulata* does not include the West Atlantic. Thompson [40] noted this species as rare or uncommon for the British fauna, occurring in the shallow water and even in intertidal mud. According to Høisæter [25], the maximum number of specimens (about 50 specimens per a sample of *P. denticulata*) was collected at 18–20 m depth. In Denmark, this species was one of the commonest members of meiobenthos whose maximum densities reached 2.230 individuals per m$^2$ [24]. In the Barents Sea, *P. denticulata* inhabits depths of 27–68 m and is numerous there [26]. Our data demonstrate a wider bathymetric range of this species' habitat in the Barents Sea: from 2 to 122 m in depth, and from 3 to 343 specimens/m$^2$ in density (Table 1). *Philinissima denticulata* is recorded for the first time from the White Sea. In this area specimens were collected at depth of 2–9 m in sediments rich with detritus.

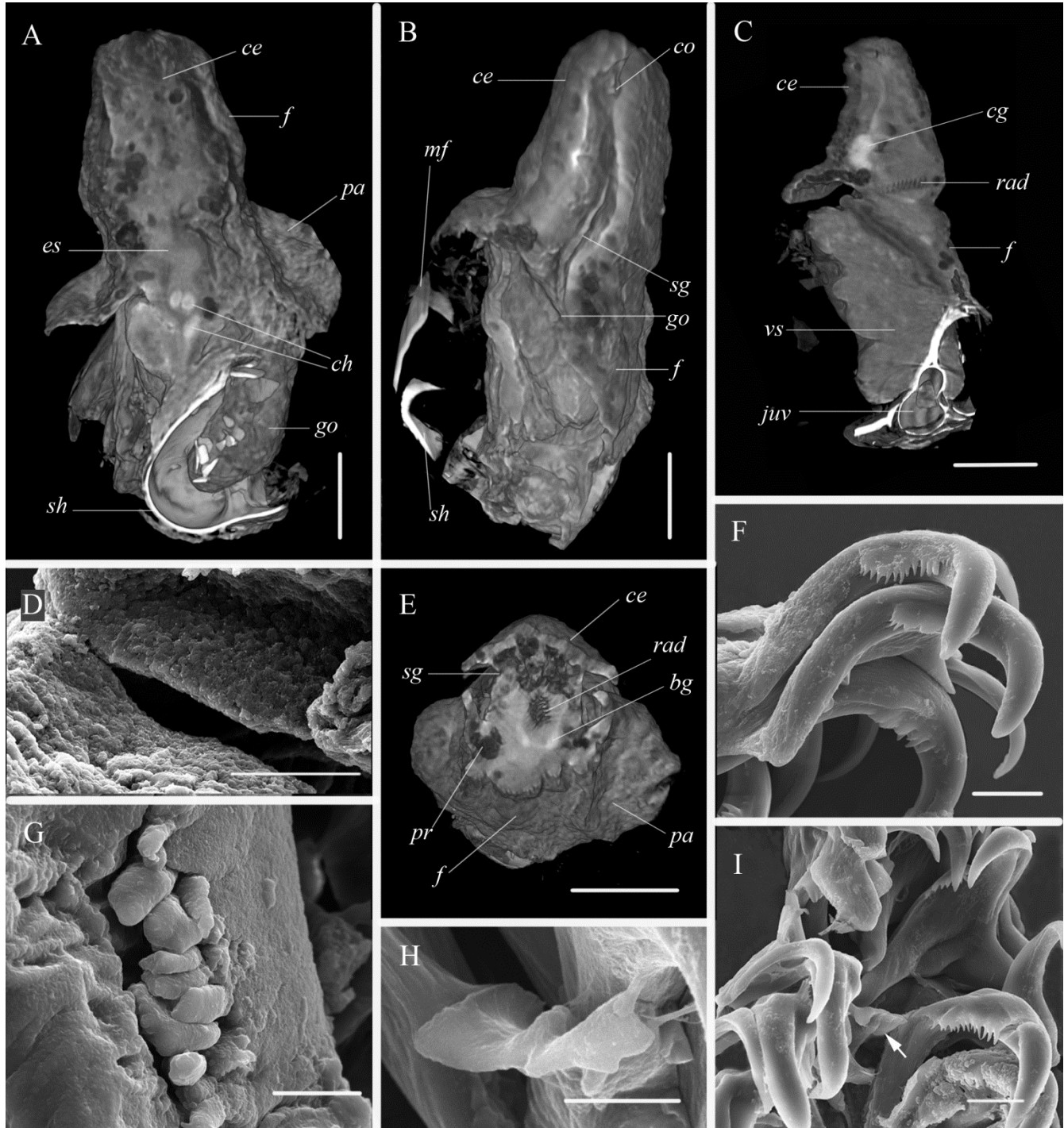

**Figure 4.** *Philinissima denticulata* specimen from the White Sea ((**A–C,E**), mCT) and specimen from the Barents Sea ((**D,F–J**), SEM), internal morphology. (**A**)—specimen, dorsal view; (**B**)—specimen, lateral view; (**C**)—specimen, optical section through radula region, lateral view; (**D**)—chitin ridges of gizzard; (**E**)—specimen, optical section through radula region, frontal view; (**F**)—radula, lateral teeth of right side; (**G**)—jaws; (**H**)—radula, rachidian tooth enlarged; (**I**)—radula, rachidian teeth marked by arrow. Abbreviations: bg—buccal ganglia, ce—cephalic shield; cg—cerebro-pedal ganglia, ch—chitinous zones of gizzard, co—copulatory system opening; es—esophagus, f—foot; go—gonad, juv—juvenile shell, mf—mantle fold; pa—parapodia, pr—prostate, sg—external seminal groove; sh—shell, vs—visceral sac. Scale bars: 5 μm (**H**), 10 μm (**D,F,G,I**), and 0.5 mm (**A–C,E**).

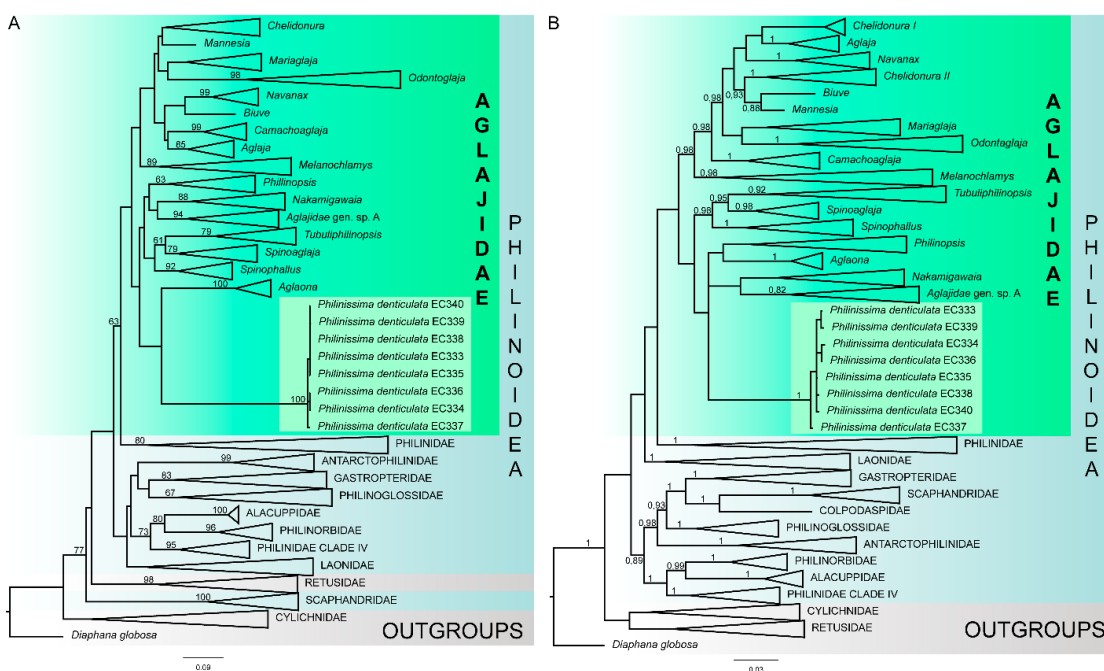

**Figure 5.** Molecular phylogenetic reconstructions based on a concatenated dataset of four genes (COI + 16S + H3 + 28S). Outgroup clades and genus-level clades within Aglajidae are collapsed. (**A**)—Maximum likelihood tree, numbers above branches indicate bootstrap support (only values higher than 60 are shown). (**B**)—Bayesian tree, numbers above branches indicate posterior probabilities (only values higher than 0.8 are shown).

*Biology.* We were able to study in vivo specimens of *Philinissima denticulata* collected in the White Sea. Animals were collected on 20 June 2022 at a depth of 8–9 m. They had mature eggs which were clearly seen through the semitransparent shell. Some specimens were kept in lab conditions in a fridge at +9 °C (average temperature of water surface in the White Sea in June) for several days. Spawning occurred after one day. The egg masses are short and oval in outline, and slightly more than 1 mm in diameter (Figure 2J). They are white in color and have a proximal tail-like brownish protuberance for anchoring in the ground. The egg capsules were tied together with a mucous string. The strings of egg capsules formed numerous regular coils within the clear gelatinous envelop. Each capsule contained one ovum. Embryos reached the earliest stage of veliger on the fourth day. At this stage they already had the larval kidney of black color, and rotated inside the capsule. The hatching of veligers started on the fifth day. The shell of the veligers consisted of 3/4 whorls. We had no opportunity to study further development of the veligers.

*Remarks. Philinissima denticulata* was regarded previously as *Philine denticulata* [12,23–26,40–42]. Pilsbry [43], Marshall [44], and Pruvot-Fol [45] regarded the species as *Philine nitida* Jeffreys, 1867. However, *P. nitida* is a junior synonym of *P. denticulata* (see Ohnheiser and Malaquias [12]).

*Philinissima denticulata* is easily distinguished by its wide cylindrical shell partly covered by the mantle. For the first time, we report the presence of the rachidian tooth in the radula of *P. denticulata*. This tooth is small and is not seen in each radula row. Possibly, the rachidian teeth of the species can be easily lost, as known for the genus *Scaphander* [46,47]. An additional inspection of previously published images of *P. denticulata* radula published by Ohnheiser and Malaquias [12] (Figure 5B in the referred paper) indicated the presence of the rachidian teeth, which was likely overlooked by the authors due to a small size of these teeth and their low number. Same rachidian teeth also may be found in the radula of the Northeast Pacific species "*Philine*" *baxteri* Valdés, Cadien & Gosliner, 2015 (see Valdés et al. [15] Figure 15G in the referred paper, and Table 3). Apart of Scaphandridae, in Philinoidea, only members of families Alacuppidae, Philinoglossidae and Antarctophilinidae have rachidian teeth.

**Table 3.** Comparative morphology of species of the families Philinidae, Laonidae, and Aglajidae with developed shell and without gizzard plates.

| Species | Shell | Penis | Gizzard Plates | Radula | Current Family | References |
|---------|-------|-------|----------------|--------|----------------|------------|
| *Philine infortunata* Pilsbry, 1895 | external, globose | no data | no data | no data | Philinidae | [12] |
| *"Philine" hemphilli* Dall, 1919 | presumably internal, globose | no data | absent | 5:1:0:1:5 | Philinidae | "may be a member of *Laona*" [15] |
| *"Philine" bakeri* Dall, 1919 | ?internal, ovate; sculpture chain-like, callus "wing-like" | no data | absent | 2:1:0:1:2 | Philinidae | [15] |
| *"Philine" rubrata* Gosliner, 1988 | internal, bulloid; sculpture punctate | prostate simple | chitinous ridges | 2:1:0:1:2 | Philinidae | [48] |
| *"Philine" baxteri* Valdés, Cadien & Gosliner, 2016 | internal, bulloid: sculpture punctuate, callus "wing-like" | prostate simple | absent | 2:1:1:1:2 | Philinidae | [15], this study |
| *Philinissima denticulata* (J. Adams, 1800) comb. nov. | external, cylindrical; sculpture fine net-like | prostate short, simple; penial papilla unarmed | chitinous ridges | 1:1:1:1:1 | Aglajidae | [12], this study |
| *Aglaona rudmani* Chaban et al., 2022 | internal, bulloid, oval: sculpture chain-like | prostate short; penial papilla armed | absent | 2:1:0:1:2 | Aglajidae | [28] |
| *A. valdesi* Chaban et al., 2022 | internal, bulloid, oval; sculpture chain-like | prostate short; penial papilla armed | absent | 2:1:0:1:2 | Aglajidae | [28] |
| *Laona zonata* A. Adams, 1865 | internal, ovate, with brown band; sculpture net-like | no data | no data | no data | Laonidae | [49] |
| *L. confusa* (Ohnheiser & Malaquias, 2013) | partly external, globose, smooth | prostate simple | absent | 3:1:?:1:3 | Laonidae | [12,18] |
| *Laona nanseni* Malaquias, Ohnheiser, Oskars & Willanssen, 2016 | external, oval, with faint rugose sculpture | flat tube | absent | absent | Laonidae | [16] |
| *"Philine" retifera* (Forbes, 1844) | external, elongate; sculpture with reticulate pattern | no data | no data | no data | *insertae sedis* | [12,16] as *Philine retifera* (Forbes, 1844) *insertae sedis* |

The development of *P. denticulata* was previously provided in detail in the excellent paper of Horikoshi [24] for specimens from Denmark. In this case, the size of egg masses was 1–1.5 mm, the size of newly hatched larvae was 90 × 70 μm, and the shell size was 30 × 95 μm. Our specimens spawn similar egg masses in both form and size, but the shells of newly hatched veligers were significantly bigger—80 × 125 μm. It is not clear whether this difference is a result of different conditions of the spawning and development of *P. denticulata* (temperature, salinity) or if there are two cryptic species—further study is needed to address this question.

*3.2. Molecular Analysis*

Single-gene trees (Supplementary Data S1) were poorly resolved at the genus- and family-levels, while the concatenated trees provided better resolution (Figure 5). The topology and node support of the concatenated trees reconstructed with Bayesian Inference (BI) and the maximum likelihood analysis (ML) were mostly congruent with few exceptions (see below). In both analyses, *Philine denticulata* appeared within the family Aglajidae, apart from the monophyletic Philinidae *s.str.* (Figure 5). However, the monophyly of the family Aglajidae received no statistical support in both ML and BI (posterior probabilities from BI (PP) = 0.69; bootstrap support from ML (BS) = 44). In addition, the position of *P. denticulata* was different: it either grouped with the deep-sea *Algaona*, or formed an unresolved polytomy with *Aglaona, Philinopsis,* and *Nakamigawaia* (Figure 5). In both cases, these relationships were not supported. The deep relationships within Aglajidae were not supported either. An additional analysis with separate partitions for different codon positions in COI revealed similar results, but bootstrap support of ML and posterior probabilities of BI were slightly higher (Figures S3 and S4). Overall, our results showed high molecular divergence of *P. denticulata* which is not related to the family Philinidae *s.str.* but has a closer position to the family Aglajidae. Therefore, we suggest establishing the new genus *Philinissima* gen. nov. and treating it as a member of the family Aglajidae until the resolution of the molecular reconstruction is improved with comprehensive taxon sampling.

## 4. Discussion

### 4.1. Molecular Data

Despite previous studies that considered *P. denticulata* to be a member of the family Philinidae [12], our results indicate that this species appears in a single clade with the genus *Aglaona* (Maximum likelihood analysis, Figure 5A), and this clade is a member of the large and diverse family Aglajidae. In Bayesian analysis, this species also nests within Aglajidae, but has unresolved relationships with *Aglaona, Nakamigawaia,* and *Philinopsis.* In both cases, these relationships are not supported statistically as posterior probabilities, and the bootstrap support is very low. Additional phylogenetic reconstructions considering possible saturation in the COI marker (Figures S3 and S4) show similar results and higher support of deep relationships within Aglajidae; however, these values are still too low to support possible inter-generic relationships. In any case, our results show that *P. denticulata* represents a new genus which forms a distinct derived lineage outside Philinidae *s.str.* and is closely related to the family Aglajidae. This case is similar to the previously reported situation with the genus *Aglaona*, whose representatives were initially identified as members of the family Laonidae based on their morphology [19]. Only with the addition of molecular genetic data, their actual phylogenetic position became evident [19,28]. According to the existing evidence, the new genus *Philinissima* should be considered a part of the family Aglajidae. However, as this is a second case of an unexpected positioning of a cephalaspid at the basal position inside Aglajidae, it may be an indication of overlooked diversity. Additionally, low node supports within Aglajidae in contrast to previous studies [50] also indicate an unstable position of both *Aglaona* and *Philinissima* on the tree, which is likely to be reconsidered in further studies of Philinoidea.

As Cephalaspidea diversity still remains understudied, and many groups yet lack molecular data, two putative outcomes are possible. Either we are considerably misguided

in the Aglajidae diagnosis, and some other animals belonging to the group could be currently wrongly taxonomically placed elsewhere. Thus, further re-description of the family will be needed upon recovery of further groups of cephalaspids as members of Aglajidae clade. Or, if more specimens are found to belong specifically to the *Aglaona/Philinissima* clade, and this group recover as separated from the Aglajidae, a new family should be described for this clade, and the Aglajid diagnosis could be reverted to a stricter form, as it was before the *Aglaona* discovery.

### 4.2. Taxonomy

*Philinissima denticulata* was originally described from the North Sea (The Wash, England) [51]. The species was collected in the East Atlantic from England to Norway, from Tunisia in the Mediterranean Sea and Denmark in the Baltic Sea, in the Barents Sea [24–26,40,42,44] and the White Sea (this study) in the Arctic. It is an abundant species found in boreal meiobenthic communities of the North Atlantic, the Barents [24,26] and White Seas (this study). The penetration of this boreal species into the Arctic is likely facilitated by the warm North Cape Current. This mollusk inhabits relatively shallow depths (from 8 to 122 m) in the Barents and White Seas. It breeds actively in June at a water temperature of +9 °C and salinity of 26‰. However, the bathymetric range of *P. denticulata* is quite large, from intertidal zone in North Wales [40] to 183 m in the Mediterranean Sea (Tunisia, see Marshall [44]), and even to 1262 m on the slope of British Channel [44]. We presume that sampling from different parts of the distribution and depth range, suitable for molecular analysis, can possibly uncover cryptic or pseudocryptic species within the genus *Philinissima*. This assumption is based on the differences in the size of newly hatched veligers from the population of the White Sea (this study) and the population from the Baltic Sea (Øresund, see Horikoshi [24]).

Currently, not all genus-level taxa which are considered synonyms of the genera *Philine* or *Laona* A. Adams, 1865 have been studied by molecular phylogenetic methods. We compared the morphology of such taxa with *Philinissima* to exclude possible synonymy. These taxa are *"Philine" hemphilli* Dall, 1919 (the type species of *Woodbridgea* Berry, 1953), *Laona zonata* A. Adams, 1865 (the type species of *Laona*), and *"Philine" retifera* (Forbes, 1844) (the type species of *Johania* Monterosato, 1884). The exotic *"Philine" retifera* (for which the internal morphology is not known, see Table 3) has an openwork deep reticulated sculpture [16] (see Figure 8A–D in the referred paper). *Laona zonata* (data on its internal morphology are also absent) has an internal bulloid shell with a brown band and reticulate sculpture [50] and differs significantly from *Philinissima denticulata*. *"Philine" hemphilli* (Table 3) from the Northeast Pacific has a globose shell ("presumably internal"—see Valdés et al. [15]) and the radula with 5 marginal teeth [15] (see Figure 12I in the referred paper). Therefore, all three species differ from *Philinissima denticulata*, and there is no evidence that *Woodbridgea*, *Laona*, and *Johania* are possible synonyms of the *Philinissima* gen. n.

Aglajids commonly inhabit tidal and upper subtidal waters in tropical and temperate regions [52–55]. A limited number of species are known from boreal and subtropical waters of the North Pacific, where they are found in shallow or subtidal waters down to 235 m [48,56,57]. Recently, unusual aglajids of the genus *Aglaona* Chaban, Ekimova, Schepetov & Chernyshev, 2022 were described from the abyssal zone of the North Pacific at a depth of 3206–3580 m [28]. Both genera—*Philinissima*, inhabiting the Arctic, and *Aglaona*, found in the Sea of Okhotsk and the Kuril–Kamchatka Trench—retained a number of plesiomorphic characters. The genus *Philinissima* gen. n. has plesiomorphic characters such as an external relatively well-developed shell, a simple penis with a short prostate, unarmed penial papilla, and the radula with rachidian teeth. These structures are not characteristics for the traditional Aglajidae. Their shell is greatly reduced and is completely covered by the mantle; the radula is absent in the most of the genera of the family, except *Odontoglaja* Rudman, 1978 and *Mannesia* Zamora-Silva & Malaquias, 2017; penis is armed with chitinous stylet [48,56,58,59]. *Philinissima* is morphologically quite similar to the genus *Aglaona*. Both genera have a well-developed shell and the radula. However, there are a

number of differences. The shell of *Aglaona* is internal, the gizzard lacks chitinous ridges, penis is armed with chitinous plate or stylet, the radula lacks rachidian tooth [28]. These two groups are found at the fringe of distribution of the family, where they successfully coexist with deep-water Philinidae in the North Pacific and shallow-water Laonidae in the Arctic. Both of these finds give us a better basis for reconstructing the morphology of the aglajid most recent common ancestor.

We expect that more taxa (in addition to *Aglaona* and *Philinissima denticulata*) close to Aglajidae *s.str.* will be found in the future. In this regard, it is interesting to note three species. *"Philine" rubrata* Gosliner, 1988 (Table 3) is now considered to belong to the genus *Philine* (Philinidae). Its gizzard bears chitinous ridges similarly to *Philinissima denticulata*. Moreover, its body is brightly colored [50,60], which is one of the typical aglajid features. Another interesting pair of species is *"Philine" bakeri* (Dall, 1919) and *"Philine" baxteri* Valdés, Cadien & Gosliner, 2016. It was suggested previously that *"Philine" bakeri* may represent a taxon closely related to the deep-sea *Aglaona* because of the high similarity of shell morphology in these mollusks [28]. The "wing-like" form of the callus of the shell is one of the diagnostic characters of the genus *Aglaona* [28]. Therefore, it is assumed that *"Philine" bakeri* can also belong or be sister to the genus *Aglaona* and to other "unusual" aglajids such as *Philinissima*. The same is true for *"Philine" baxteri* having a "wing-like" callus [15] (see Figure 15B in the referred paper) and rachidian teeth [15] (see Figure 15G in the referred paper) similar to those of *Philine denticulata*.

An internal shell completely covered by the mantle is found in most Philinoidea species. In this regard, it is quite interesting to note that *Philine infortunata* Pilsbry, 1895 (with unknown details in morphology, currently considered as a member of the family Philinidae), *Laona nanseni* Malaquias, Ohnheiser, Oskars & Willassen, 2016 (Laonidae), and *"Philine" retifera insertae sedis* have an external shell. In *Laona confusa* (Ohnheiser & Malaquias, 2013) (Laonidae) and *Philinissima denticulata* (Aglajidae) the shell is partly external. This implicates that shell internalization independently occurred in different Philinoidea lineages, and, in contemporary Aglajidae, a full spectrum of reduction can be found from a completely developed external shell to a greatly reduced internal one.

**Supplementary Materials:** The following supporting information can be downloaded at: https://www.mdpi.com/article/10.3390/d15030395/s1, Figure S1: COI saturation plot. Figure S2: H3 saturation plot. Figure S3: maximum likelihood phylogenetic tree based on the four-gene dataset with COI separated to three partitions by a codon position, numbers above branches indicate bootstrap support. Figure S4: Bayesian phylogenetic tree based on the four-gene dataset with COI separated to three partitions by a codon position, numbers above branches indicate posterior probabilities. Data S1: unedited molecular phylogenetic trees based on single-gene markers, maximum likelihood analysis.

**Author Contributions:** Conceptualization, E.C. and I.E.; methodology, E.C., I.E., E.N. and D.S.; software, I.E., E.N. and D.S.; validation, E.C., I.E. and D.S.; formal analysis, E.C., I.E., E.N. and D.S.; investigation, E.C. and I.E.; resources, E.C. and P.L.; data curation, E.C., I.E. and D.S.; writing—original draft preparation, E.C., I.E. and D.S.; writing—review and editing, E.C., I.E., P.L., E.N. and D.S.; visualization, E.C., I.E., E.N. and D.S.; supervision, E.C. and I.E.; project administration, E.C. and I.E.; funding acquisition, I.E. All authors have read and agreed to the published version of the manuscript.

**Funding:** This study was carried out within the framework of the Ministry of Education and Science of the Russian Federation no. 1021051402797–9 (for E.C.) with financial support of the Russian Science Foundation grant no. 20–74–10012 to E.C., I.E., E.N. and D.S.

**Institutional Review Board Statement:** Not applicable.

**Informed Consent Statement:** Not applicable.

**Data Availability Statement:** Unedited trees are provided as Supplementary Materials. All sequences are deposited to the GenBank.

**Acknowledgments:** We are deeply grateful to Alexander Chaban (St. Petersburg) and Anna Zhadan (Moscow State University) for their assistance in obtaining a live specimen from the White Sea; Alexander Tsetlin for the opportunity to collect and study material at the White Sea Biological Station of the Moscow State University (WSBS, MSU). Special thanks are also due to Alexey Mirolubov (Zoological Institute, St. Petersburg) for the arrangement of the SEM facilities at ZIN RAS, Ludmila Flyachinskaya (ZIN RAS) for the assistance with the photos of the live specimen, Valentina Tambovtseva (N.K. Koltzov Institute of Developmental Biology RAS, Moscow) and Maria Stanovova (MSU) for the assistance in Sanger sequencing, Andrey Lavrov and Fedor Bolshakov for the assistance in the Center of Microscopy, WSBS MSU. Molecular study was conducted using the equipment of the molecular lab at the Invertebrate Zoology Department, MSU and of the Core Centrum of the N.K. Koltzov Institute of Developmental Biology, RAS. Morphological analysis was partly carried out using equipment of the Center of Microscopy, WSBS MSU. Scientific research was conducted using equipment of the "Taxon" Research Resource Center of ZIN RAS.

**Conflicts of Interest:** The authors declare no conflict of interest. The funders had no role in the design of the study, in the collection, analyses, or interpretation of data, in the writing of the manuscript, or in the decision to publish the results.

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
