# Peer review of "Bizarre Morphology Obscures Real Affiliation: An Integrative Study of Enigmatic Cephalaspid Philine denticulata from Arctic Waters Reveals Its Unique Phylogenetic Position"

_diversity, doi:10.3390/d15030395_

Round 1

Reviewer 1 Report

Dear colleagues, you have performed another taxonomic study of your mollusks and, as it seems, once again made a good discovery. Nevertheless, I ask you to present the procedure and results of the molecular phylogenetic analysis in more detail, so that no one will doubt the conclusions. You illustrate results solely by phylogenies based on a concatenated dataset of four markers, COI, 16S, H3, and 28S. But you didn't study 16S and it's unclear how you could come up with it for your individuals using unnamed “GenBank data” (L 112-113). I suggest you show your empirical data, including phylogenies for each gene individually. If you insist on including 16S in a complex analysis, explain in details the rationale and procedure.

Minor comments

Figure 1. - Atlantic Ocean

93. The authors' contributions are revealed in a special section, aren't they?

128-129. Sequences were concatenated by a simple biopython script (19) - Please check the reference, it is probably not correct.

368. Do as you like, but Table 3 would look better in Results than in Discussion.

445. Data S1: unedited molecular phylogenetic trees based on four markers. - I could not find this file in the appendix diversity-2166166-supplementary.zip.

Author Response

Thank you very much for a high evaluation of our work and useful comments.

Regarding the molecular analysis, the 16S is a standard marker for phylogenetic studies of Cephalaspidea. We failed to amplify this marker for D. denticulata likely due to the divergence in DNA binding site of standard 16S primers. However, for the better resolution of trees at the genus- and the family-levels we decided to incorporate this marker into the phylogenetic analysis, although data for P. denticulata were marked as missing.

We added Table S1 with NCBI accession numbers of all sequences used in the phylogenetic reconstruction.

We also provided four trees based on the single-gene analyses (Data S1).

Corresponding parts of the manuscript were corrected accordingly.

Minor comments

2. Figure 1. – Atlantic Ocean

Corrected

3. 93. The authors’ contributions are revealed in a species section, aren’t they?

Yes, the name was deleted

4. 128-129. Sequences were concatenated by a simple biopython script (19) – Please check the reference, it is probably not correct.

The reference is correct; in Chaban et al., 2019 this biopython script is applied for the first time and available as Supplementary material.

5. 368. Do as you like, but Table 3 would look better in Results than in Discussion.

-We agree. The Table 3 was moved in Results.

6. 445. Data Sq: unedited molecular phylogenetic trees based on four markers. – I could not find this file in the appendix diversity-2166166.zip.

-Unedited phylogenetic trees based on four markers are Figures S3&S4, Data S1 with single gene trees was included in the Supplementary material

Reviewer 2 Report

Authors  provided  morphology and molecular data for the close relationships of the species Philine denticulata to the family Aglajidae. The data were full and accurate. I suggested for publication after some minor  revision.

1, the data present sequence should modified. The morphology of Philine denticulata from the White Sea and the Barents Sea using the light microscopy and the scanning electron microscopy, and computer microtomography should firsted presented. And then the molecular data should be introduced.

2, Molecular phylogentic reconstructions were the results of four genes. Were there some differences between different genes?

Author Response

1. the data present sequence should modified. The morphology of Philine from the White Sea and Barents Sea using the light microscopy and the scanning electron microscopy, and computer microtomography should firsted presented. And the molecular data should be introduced.

-Done

2. Molecular phylogenetic reconstructions were the results of four genes. Were there some differences between different genes?

No, the single-gene trees were poorly resolved at the genus- and family-levels and therefore they did not contradict the concatenated tree. We added a couple of sentences in the M&Ms and Results sections, and also provide single-gene trees as Supplementary material Data S1.

Reviewer 3 Report

Overall this is a sound integrative study, its conclusions are supported by both an impressive morphological and a (somewhat less impressive) molecular dataset. The authors place their findings into context and I think that their careful consideration of taxonomic options has a lot of merit. They acknowledge the uncertainty regarding the molecular position of their target species and I think its assignment as a new genus that is tentatively placed in the Aglajiidae is sensible.

However, I do have a few points that I think should definitely be addressed before this manuscript is accepted for publication:

1. Methods - The methods section is generally fine. I am missing two things, though. Firstly, the total length of the concatenated dataset is provided in the results, but this includes the 16S fragment that the authors failed to amplify for P. denticulata. It is fine to refer to previously published protocols, but please state how many bp you obtained for the three genes you sequenced for that species. Secondly, more importantly, the methods section does not mention outgroups. From Fig. 2 it seems that all taxa outside Aglajiidae including P. reticulata were used as outgroups? If you want to test whether P. reticulata is a philinid or not, you should not include Philinidae among your outgroups. Please re-root your tree or simply correct Fig. 2, in case you did not root the topology with all the taxa labelled outgroups.

2. Language: the MS can easily be understood, but the grammar of some sections, not least the abstract, is rather poor. If possible, let a native speaker (it should be a biologist!) proofread the MS. I did not feel it to be my job to improve it linguistically. There are also a few obvious mistakes (gas net (l. 102) = gauze net) or riches (l. 294) = reaches.

3. Minor stuff: Fig. 1 - „decimal degrees“ – this is not what you are showing and irrelevant, delete for both latitude and longitude in both panels.

Author Response

1. Methods – The methods section is generally fine. I am missing two things, though.

Firstly, the total length of the concatenated dataset is provided in the results, but this includes the 16S fragment that the authors failed to amplify for P. denticulata. It is fine to refer to previously published protocols, but please state how many bp you obtained for the three genes you sequenced for that species.

The length of alignments for each locus is now given in the M&Ms section.

Secondly, more importantly, the methods section does not mention outgroups. From Fig. 2 it seems that all taxa outside Aglajidae including P. reticulate were used as outgroups? If you want to test whether P. reticulate is a philinid or not, you should not include Philinidae among your outgroups. Pleas re-root your tree or simply correct Fig. 2, in case you did not root topology with all the taxa labeled outgroups.

Diaphana globosa was chosen as distant outgroup following Oskars et al. 2015, Cylichnidae and Retusidae were used as outgroups. This information was added to the M&Ms section.

We corrected the Figure 2 (now Figure 5, after the reorganization) according to the Reviewer’s suggestions.

2. Language: the MS can easily be understood, but the grammar of some sections, not least the abstract, is rather poor. If possible, let a native speaker (it should be a biologist!) proofread the MS. I did not feel it to be my job to improve it linguistically. There are also a few obvious mistakes (gas net (l.102) = gauze net) or riches (l. 294) = reaches.

We corrected the manuscript text paying special attention to abstract and taxonomy sections.

3. Minor stuff: Fig. 1 – “decimal degrees” – this is not what you are showing and irrelevant, delete for both latitude and longitude in both panels.

-Corrected

Round 2

Reviewer 1 Report

I am satisfied with the changes made. Thank you for your cooperation.